# Neuroanatomical Correlates of Semantic Features of Narrative Speech in Semantic and Logopenic Variants of Primary Progressive Aphasia

**DOI:** 10.3390/brainsci12070910

**Published:** 2022-07-12

**Authors:** Davide Quaranta, Sonia Di Tella, Camillo Marra, Simona Gaudino, Federica L’Abbate, Maria Caterina Silveri

**Affiliations:** 1Neurology Unit, Fondazione Policlinico Universitario ‘Agostino Gemelli’ IRCSS, 00168 Rome, Italy; camillo.marra@policlinicogemelli.it (C.M.); federica.labbate@guest.policlinicogemelli.it (F.L.); 2Department of Psychology, Università Cattolica del Sacro Cuore, 20123 Milan, Italy; sonia.ditella@unicatt.it (S.D.T.); mariacaterina.silveri@unicatt.it (M.C.S.); 3Department of Neuroscience, Università Cattolica del Sacro Cuore, 00168 Rome, Italy; 4Radiology and Neuroradiology Unit, Fondazione Policlinico Universitario ‘Agostino Gemelli’ IRCSS, 00168 Rome, Italy; simona.gaudino@policlinicogemelli.it; 5Centre for the Medicine of the Aging, Fondazione Policlinico Universitario ‘Agostino Gemelli’ IRCSS, 00168 Rome, Italy

**Keywords:** primary progressive aphasia, semantic memory, lexical databases, cortical thickness

## Abstract

The semantic variant of a primary progressive aphasia (svPPA) is characterized by progressive disruption of semantic knowledge. This study aimed to compare the semantic features of words produced during a narrative speech in svPPA and the logopenic variant of PPA (lvPPA) and to explore their neuroanatomical correlates. Six patients with svPPA and sixteen with lvPPA underwent narrative speech tasks. For all the content words, a semantic depth index (SDI) was determined based on the taxonomic structure of a large lexical database. Study participants underwent an MRI examination. Cortical thickness measures were extracted according to the Desikan atlas. Correlations were computed between SDI and the thickness of cortical regions. Mean SDI was lower for svPPA than for lvPPA. Correlation analyses showed a positive association between the SDI and the cortical thickness of the bilateral temporal pole, parahippocampal and entorhinal cortices, and left middle and superior temporal cortices. Disruption of semantic knowledge observed in svPPA leads to the production of generic terms in narrative speech, and the SDI may be useful for quantifying the level of semantic impairment. The measure was associated with the cortical thickness of brain regions associated with semantic memory.

## 1. Introduction

Primary progressive aphasia (PPA) is a group of neurodegenerative disorders in which the fundamental disturbance is progressive disruption of language processing. The current clinical classification includes three main variants: semantic variant (svPPA), non-fluent variant (nfvPPA), and logopenic variant (lvPPA) [1]. This classification is based on the nature of language disturbances observed in patients, supported by data obtained using neuroimaging techniques (magnetic resonance imaging [MRI] and positron emission tomography).

Progressive and multimodal loss of semantic knowledge is the main feature of the svPPA. At the disease onset, the principal disturbances are represented by anomia and difficulties in comprehending single words, which are more evident for words and concepts characterized by low familiarity and frequency [2,3]. Confrontation-naming and single-word comprehension tasks are typically used to assess these disturbances. Patients with svPPA produce semantic paraphasias, tending to substitute low-familiarity words with more familiar words (“cat” for “cheetah”) and often resorting to superordinate categories (“animal” for “cheetah”) [3]. Disease progression is paralleled by a progressive and multimodal impairment of object recognition, leading, for example, to the inability to make proper use of a tool [4]. In some patients, the main disturbance is impaired multimodal recognition [5,6]. The neuroradiological hallmark of svPPA is atrophy of the anterior part of the temporal lobe (ATL) [1,3], which is generally asymmetrical at disease onset. It is more prominent in the left temporal lobe in most patients with disorders of word and object recognition, whereas patients with unique entities recognition disorders have prominent right ATL atrophy [7]. Most patients affected by svPPA display neuropathological changes characterized by TDP-43 inclusions [8].

According to current diagnostic criteria, patients with lvPPA present with word-finding deficits and difficulties in sentence repetition, and in the absence of clear agrammatism, apraxia of speech, or semantic memory impairment [1]. However, the nosological position of lvPPA has been questioned because of the lack of distinctiveness of the clinical features required to obtain a diagnosis [9,10] and the variability of clinical features that may be observed at different time points from the diagnosis with respect to the possible onset of agrammatism [11,12,13]. However, a substantial proportion of patients with PPA has neither the semantic nor the nonfluent variants, and they do not necessarily conform to a discrete logopenic variant, leading some authors to define a mixed form of PPA [12]. One of the core deficits in lvPPA is the disorder of the phonological loop, a component of working memory responsible for the short-term representation of verbal information [14]. This impairment is considered the fundamental mechanism determining difficulty in comprehension, repetition of sentences, and phonological errors [15,16]. LvPPA is associated with regional atrophy in the posterior perisilvian and inferior parietal areas and is usually more pronounced in the left hemisphere [1,13,17]. This pattern of atrophy is consistent with the hypothesis of impairment of phonological loop functions that are thought to be elaborated in the inferior parietal cortex [18]. From a neuropathological point of view, lvPPA patients are mostly affected by Alzheimer’s type pathology [19].

In recent years, a novel approach to the diagnosis of PPA has been represented by connected-speech analysis, which allows the assessment of lexical and grammatical processing and communicative competence skills. Connected speech analysis provides rich information that enables us to examine speakers’ ability to use language in context [20], in contrast to speech elicited one word at a time, with full attention focused on the production of a single word. The most common tasks for collecting connected-speech data are represented by picture description, story narration, and different modalities of conversation [21]. The study of connected speech develops around characters and events and assesses the setting of events, related actions, and resolving themes [22].

Narrative productions can be analyzed in terms of two major levels of analysis [23,24]: (1) a microlinguistic level that focuses on lexical and grammatical processing and contributes to “intrasentential” structure (within-utterance level) and (2) a macrolinguistic level that focuses on pragmatic and discourse-level processing, responsible for “intersentential” organization (between-utterance level) [22,25]. The generation of well-structured and informative narratives relies on both lexical and grammatical skills and the capability to establish accurate, cohesive, and coherent links among utterances, as well as integrate sentential meanings with a linguistic and extra-linguistic context.

Neuroimaging studies have suggested that narrative coherence may depend on a network involving the bilateral medial, lateral frontal, and anterior temporal regions and the precuneus [26,27]. A study by Troiani and colleagues [28] reported that narrative production activates the inferior frontal cortex bilaterally and the left dorsal inferior-frontal and left lateral temporal-parietal regions.

Narrative skills were also investigated in patients with PPA. In lvPPA, a low speech rate and an increased number of filled pauses and false starts have been reported [15,26,29]. People with lvPPA show impairment at both phonological and syntactic levels; in particular, phonemic errors represent an important facet of this variant [26]. Furthermore, a reduced number of open-class words were observed [30]. Notwithstanding that a core diagnostic feature of lvPPA is the absence of agrammatism, an impairment at the syntactic level has been reported by the reduced proportion of well-formed sentences [26,30].

In svPPA, a reduced number of nouns has been reported [15], with some studies showing an increased frequency of nouns and verbs produced [21]. Syntactic processing is generally intact; however, some analyses have shown that the proportion of well-formed sentences is reduced compared with that in normal individuals [31]. As is easily predictable, semantic errors are a constant finding in these patients [32,33]. However, the typical pattern of semantic errors described in svPPA, that is, the production of superordinate words and generic terms instead of the requested terms, has generally been described using single-word production (for example, in confrontation naming) and comprehension.

In this study, we aimed to compare the microlinguistic features, and particularly the semantic features, of words produced during a narrative speech in svPPA and lvPPA and to explore their neuroanatomical correlates.

## 2. Materials and Methods

### 2.1. Participants

We recruited 22 participants (16 lvPPA patients; 6 svPPA patients) at the Center for the Medicine of the Aging of the Department of Geriatrics, Neurology, and Orthopedics of the Catholic University of Rome between 2013 and 2014. Aphasic patients fulfilled the criteria for both lvPPA and svPPA [1]. All the patients were native Italian speakers. They underwent a brain MRI scan within a maximum interval of two months from the neuropsychological examination (i.e., either before or after). Patients showed relatively preserved autonomy in activities of daily living and were included only if they had at least five years of formal education, did not express themselves in dialect, and completed the neuropsychological examination. We excluded patients with a history or radiological evidence of cerebrovascular disease, a history of major psychiatric disorders, or alcohol or drug abuse. Moreover, we excluded patients with visual-perceptual difficulties that could interfere with the perception of the stimulus picture.

All participants were right-handed, according to the Edinburgh Handedness Inventory. All participants provided written informed consent to participate in the study after all procedures were fully explained. This study was approved by the Catholic University Institutional Ethics Committee.

### 2.2. Neuropsychological Assessment

The cognitive profiles of all patients were carefully evaluated. They underwent a full neuropsychological examination, which included tasks of verbal episodic memory, short-term memory, visuospatial analysis, constructive praxis, and intelligence. These cognitive functions are required to complete a narrative task. The complete battery took 45–60 min (in one or two sessions) to administer. The general neuropsychological assessment included memory tasks (Rey’s immediate and delayed free recall of 15 words, Rey Osterrieth figure delayed recall), visuospatial tasks (line cancellations), phonological and semantic fluency, Raven’s colored matrices, digits, and spatial span.

Language was investigated using Capasso and Miceli’s [34] Esame neuropsicologico per l’Afasia-ENPA (Neuropsychological Examination for Aphasia), evaluating both the lexical and the sublexical mechanisms in production and comprehension.

### 2.3. Assessment of Narrative Abilities

Narrative ability was assessed using pictures and story descriptions. Each participant was asked to produce a set of narratives elicited with the help of two pictures, the Cookie Theft Scene (Boston Diagnostic Aphasia Examination) and the Picnic Scene (Western Aphasia Battery) and to talk about his or her illness and typical day. The Cookie Theft Scene is a black-and-white line drawing of an animated kitchen. In the picture, a woman is standing at the kitchen sink, drying a plate while the sink is overflowing with water. Meanwhile, a boy (presumably the woman’s son) has climbed a stool that is rocking unsteadily and is cooking from a jar in the cupboard. A girl (presumably the boy’s sister) has her hand raised upward to receive the cookies that her brother (the boy) is taking.

In the Picnic Scene, there is a man and a woman having a picnic near a lake. The man is reading a book, and the woman is pouring a drink into a glass. In this scene, another man is flying a kite. A dog is standing next to him.

To avoid poor performance due to short-term memory limitations, the pictures remained visible until the patient had finished his or her description. The examiners minimized prompts to allow uninterrupted speech, although a prompt was offered after 10 s of silence.

To better assess language production, a short informal conversation was reproduced. It starts with an open-ended question (i.e., “*Tell me about your family*”) to introduce a topic and encourage conversation. Participants were free to answer as they liked in order to reproduce a spontaneous conversation about common events in daily life (family, hobbies, career).

Each storytelling and conversation was tape-recorded and subsequently transcribed verbatim by an examiner. The transcription included phonological fillers, pauses, false starts, and extraneous utterances.

The close transcription of the speech samples requires approximately an hour. Speech errors were classified as phonological (phoneme substitution, deletion, or insertion) or semantic paraphasia (when a target word is substituted by a semantically related word). Moreover, we considered the number of well-formed sentences, which are complete sentences free of grammatical errors, regardless of content and the frequency of anomic pauses and conduites d’approche (CDA) in the speech. Each word produced by the patients was transcribed in a database and analyzed for the number of times it was repeated (number of occurrences) and, for open-class words, lexicosemantic features that included frequency of use (FU) [35] and semantic depth index (SDI).

The SDI was derived based on the taxonomic structure of the large lexical database WordNet, an English database developed at Princeton University [36]. WordNet is organized in a hierarchical fashion in which lexical and conceptual links connect words and groups of words. The hierarchy is organized starting from generic terms (the most generic being “entity”) towards terms that become more and more specific (see Figure 1 for an example). The fundamental element of WordNet is the synset (set of synonyms), which is defined by gloss, a verbal description of the concept to which it refers. Several conceptual relationships are considered in WordNet. For example, nouns are organized according to hypernyms, hyponyms, meronyms, and holonym relations. Hypernym and hyponym refer to the taxonomic “is-a-kind-of” relationship (if “cookie” is a kind of “food”, then “cookie” is a hyponym of “food” and “food” is a hypernym of “cookie”). Meronyms and holonyms are based on the “is-a-part-of” relationship (if “wheel” is a part of “bicycle,” then “wheel” is a meronym of “bicycle,” and “bicycle” is a holonym of “wheel” wheel’). Each synset was embedded according to an ordered hierarchy constructed on the “is a” relationship [37], resembling the categorical organization of conceptual knowledge. From a general point of view, WordNet can be regarded as a nodes-and-paths network, in which nodes are represented by synsets and paths by the connections between synsets. The SDI was determined, for each open-class word, by computing the number of nodes separating the produced word from the root term (for example, “entity” in the case of nouns). Thus, higher values corresponded to a deeper exploration of the lexical-semantic system to produce a more precise lexical entry, whereas lower values indicated the production of terms closer to the root, thus being more generic. Figure 1 shows an example of SDI computation. We report, as an example, part of a svPPA speech: “*l’acqua, non lo so se era chiuso oppure o oppure c’erano delle cose …*” (translation: the water, I don’t know or or..or if it was closed or there were things: SDI_water_: 6; SDI_thing_: 1; SDI_close_: 1). In a case of lvPPA, we recorded the following speech: “*c’è il … i vari sportelli e l’acqua che esce dal lavandino da sola*” (translation: there is the … the various doors and the water coming out of the sink on its own; SDI_water_: 6; SDI_door_: 7; SDI_sink_: 8; SDI_come out_: 3).

Since the narrative speech taken into account was in Italian, in the present study, we referred to MultiWordNet [38], a multilingual lexical database whose taxonomy is aligned to that of WordNet, which proved to be reliable in previous studies on semantic features of words produced in fluency tasks [39].

### 2.4. MRI Acquisition and Processing

All participants (except one patient with lvPPA) underwent a 1.5-T MRI (Signa, GE Healthcare) with an eight-channel phased-array neurovascular coil to obtain diffusion-weighted imaging, fluid-attenuated inversion recovery, turbo spin echo, and gradient echo images. Cortical thickness analysis was performed for all PPA participants within two months (before or after) the neuropsychological examination using a T1-weighted fast gradient echo in the sagittal plane for multiplanar reconstruction (TR = 2000 ms, TE = 3.42 ms, flip angle = 15°, FOV = 256 mm, 1.0-mm slice thickness for a total of 160 slices per slab, matrix size = 256, NEX = 1). The raw 3D T1 MRI data underwent automated processing for surface-based cortex reconstruction and volumetric segmentation using the FreeSurfer image analysis software (version 6.0.0, Martinos Center for Biomedical Imaging, Charlestown, Massachusetts), which is documented and freely available for download online (http://surfer.nmr.mgh.harvard.edu/, accessed on 14 June 2022). Briefly, after brain extraction, high-resolution 3D T1 images were registered to the standard space, and the gray–white and gray–cerebrospinal fluid borders were computed [40,41,42]. An experienced neuroradiologist visually inspected the image outputs from each stage of FreeSurfer processing and edited them to refine the segmentation and correct any software delineation errors. Cortical parcellations were made according to Desikan’s atlas [42], and the cortical thickness was extracted for all regions included in the atlas.

A set of cortical areas previously associated with concept neural representation [43,44] was selected as part of the semantic network, including the following regions of interest (ROIs): the anterior temporal lobe, superior temporal gyrus, middle temporal gyrus, inferior temporal gyrus, fusiform gyrus, parahippocampal gyrus, entorhinal cortex, inferior frontal gyrus, caudal middle frontal gyrus, superior frontal gyrus, precuneus, supramarginal gyrus, posterior cingulate gyrus, and rostral anterior cingulate gyrus.

### 2.5. Statistical Analyses

Data were analyzed with Statistical Package for Social Sciences (SPSS, IBM Corporation, Armonk, NY, USA) version 24.0. The two-tailed level of statistical significance was set at *p* < 0.05. Comparisons between groups were performed using the Mann–Whitney U test. Concerning cortical thickness analyses, the two clinical groups were statistically contrasted against those of 30 Healthy Controls (HC) matched for age, sex and education recruited in the neuroimaging laboratory. Pearson’s correlation coefficients (*r*) were computed between the mean of the semantic depth index and the thickness of the ROI of the semantic network from Desikan’s atlas [45] in the entire sample of patients with lvPPA and svPPA. The results of correlation with cortical thickness were considered statistically significant at *p* < 0.05 after the false discovery rate (FDR) correction controlling procedure for multiple comparisons [46].

## 3. Results

### 3.1. Microlinguistic Features of Narrative Speech

The sociodemographic and neuropsychological features of the patients are summarized in Table 1.

Patients affected by lvPPA produced fewer open-class words (in particular, verbs and adjectives) than those affected by svPPA (*p* = 0.002, *p* = 0.006, and *p* = 0.010, respectively) (Table 2). Furthermore, the number of corrected sentences was higher for svPPA than for lvPPA (*p* < 0.001), whereas phonemic paraphasia was more common for lvPPA (*p* = 0.040). Finally, SDI was higher in the lvPPA group than in the svPPA group (*p* = 0.033). Individual SDI scores are reported in Appendix A.

### 3.2. Between-Group Comparisons on Cortical Thickness

The comparison of cortical thickness between subjects affected by svPPA and lvPPA and a group of HC is reported in Table 3.

*SvPPA* vs. *HC*: In more detail, cortical thickness was significantly reduced in patients with svPPA compared with HC in several regions covering the medial aspect of the temporal lobe (bilateral temporal pole, entorhinal cortex, parahippocampal gyrus, fusiform gyrus), the lateral aspect of the temporal lobe (bilateral banks of the superior temporal sulcus, inferior, middle, and superior temporal gyri), the frontal lobe (bilateral *pars opercularis* of the inferior frontal gyrus, left caudal and rostral middle frontal gyrus, superior frontal gyrus, frontal pole, lateral and medial orbital frontal cortex), the parietal lobe (bilateral inferior parietal cortex and left postcentral gyrus, precuneus, superior parietal cortex, and supramarginal gyrus), the occipital lobe (left cuneus), the bilateral insula and the cingulate cortex (left isthmus, posterior and anterior cingulate cortex) (all *p*_FDR_ < 0.05).

*LvPPA* vs. *HC*: Similarly, a reduction of cortical thickness was also observed in patients with lvPPA compared to HC in numerous areas of all lobes including the medial aspect of the temporal lobe (bilateral temporal pole, entorhinal cortex, fusiform gyrus, and left parahippocampal gyrus), the lateral aspect of the temporal lobe (bilateral banks of the superior temporal sulcus, inferior, middle, and superior temporal gyri and left transverse temporal cortex), the frontal lobe (bilateral *pars opercularis* of the inferior frontal gyrus, rostral and caudal middle frontal gyrus, superior frontal gyrus, precentral gyrus and paracentral lobule, and left *pars triangularis* of the inferior frontal gyrus), the parietal lobe (bilateral inferior and superior parietal cortex, postcentral gyrus, precuneus and supramarginal gyrus), the occipital lobe (bilateral lingual gyrus, lateral occipital cortex, and left cuneus), the bilateral insula and the cingulate cortex (bilateral isthmus, posterior cingulate and left anterior cingulate cortex) (all *p*_FDR_ < 0.05).

### 3.3. Correlation between the Semantic Depth Index and the Cortical Thickness

Correlational analyses between the semantic index and the ROI of the semantic network measurements indicates a left-lateralized pattern of areas (left hemisphere > right hemisphere) showing a positive association between the mean semantic depth index and the thickness of the left entorhinal cortex (*r* = 0.668, *p_FDR_* = 0.011), left middle temporal cortex (*r* = 0.558, *p_FDR_* = 0.047), left parahippocampal cortex (*r* = 0.540, *p_FDR_* = 0.047), left superior temporal cortex (*r* = 0.559, *p_FDR_* = 0.047), left temporal pole (*r* = 0.662, *p_FDR_* = 0.011), right entorhinal cortex (*r* = 0.738, *p* < 0.001), right parahippocampal cortex (*r* = 0.555, *p_FDR_* = 0.047), and right temporal pole (*r* = 0.551, *p_FDR_* = 0.047). Although non-surviving after multiple comparison correction, a trend for correlation was also observed for the left inferior temporal cortex (r = 0.504, *p*_UNC_ = 0.020) and the fusiform gyrus (r = 0.438, *p*_UNC_ = 0.047). A visual representation of the ROIs of the semantic network which significantly correlated with SDI in the whole clinical group is given in Figure 2.

## 4. Discussion

The analysis of connected speech is a knowledgeable source of information for the study of aphasic patients, and its role is becoming more relevant in the field of neurodegenerative disorders in general and PPA in particular [21].

In this study, we explored the main features of connected speech in patients affected lvPPA and svPPA and introduced the SDI, a measure of the integrity of the lexical-semantic system.

These behavioral results are consistent with those of previous studies assessing connected speech in PPA. Compared with svPPA, patients with lvPPA produced a reduced number of open-class words, as previously reported [30], and as expected, they also produced a higher number of phonemic paraphasias [26]. We did not explore the syntactic aspects of connected speech in detail; however, as previously reported, the number of correct sentences produced by lvPPA was lower than that produced by svPPA [26,30]. This finding is in contrast to the notion that the absence of agrammatism is required to diagnose lvPPA [1]. Possibly, such difficulties may reflect a strategy to cope with lexical impairment: if the patient cannot retrieve the target word from the lexicon, he or she simply skips the undergoing proposition introducing a new argument [47].

As for the svPPA group, the only score in which we observed scores lower than those obtained by the lvPPA patients was the SDI. Even if svPPA patients produced a higher number of semantic paraphasias than lvPPA patients, this difference was not statistically significant. However, the production of semantic paraphasias may depend on both the disruption of semantic memory and difficulties in accessing the lexicon [48]. Therefore, it is not surprising that patients with lvPPA may also produce such errors in narrative speech, as previously shown by other authors [29]. However, the SDI was conceived based on the typical behavior of patients affected by svPPA, which often produces semantic errors of the “superordinate” type [3,17]. From a linguistic point of view, this could be translated into a tendency to produce words that are hypernyms [37] of the target words. However, the “superordinate category” concept can be easily applied to a relatively small group of concepts, and, to our knowledge, it is generally based on *a priori definitions*. Nevertheless, hierarchical relationships may be broader than those that can be easily recognized (for example, the superordinate category “animal” for “cat”). For this reason, the SDI may be helpful in detecting the disruption of semantic memory, even in the absence of typical superordinate-type errors.

The reliability of this measure was confirmed by the results obtained by exploring its correlation with neurostructural measurements. We found evidence of a significant association between the SDI and the cortical thickness of brain regions associated with semantic memory, covering a large area in the temporal cortex (left hemisphere > right hemisphere), including the bilateral temporal pole, parahippocampal cortex, entorhinal cortex, and left middle and superior temporal cortex.

Overall, our correlational results suggest an association between SDI and the structural integrity of temporal regions implicated in the perceptual system for processing concrete concepts rather than of the verbal system implicated in the processing of abstract concepts. This is not surprising, considering that our semantic index was derived from the connected speech collected during the picture description. The lateralization of most correlations between SDI and the thickness of the left regions is consistent with left-hemispheric functional specialization for speech production.

More specifically, the correlation with the bilateral anterior temporal lobes (ATL) is in line with the hub-and-spoke model, which recognizes this area as central to semantic cognition [49,50,51,52], serving as a transmodal hub integrating information from the surrounding sensorimotor to form amodal semantic representations [53,54]. ATLs have been hypothesized to map between different modality-specific representations and to make proper generalizations based on crucial semantic relationships rather than superficial similarities in each particular domain [55]. This region is highly interconnected via interactions with other nodes or ‘spokes’ in the network [56], and its hub-like functions allow the mediation of learned associations distributed throughout the human cortex [57]. Damage to the ATL severely disrupts semantic processes, as supported by the microlinguistic analysis of our svPPA patients.

With respect to the correlation with the parahippocampal gyrus, it is important to note that this region contributes to the processing of visual and imageable spatial property knowledge during semantic tasks [58,59].

No association emerged between SDI and regions involved in semantic control system, comprising the inferior frontal gyrus (IFG), which is engaged alongside the semantic system in directing and constraining the retrieval of relevant semantic knowledge [51,60,61,62]. The lack of correlation with this region might be explained by the involvement of this area, especially in retrieving information with weak semantic relations and selecting task-relevant information [63]. On a methodological note, we acknowledge that the number of participants in this study was relatively small. Nevertheless, we acknowledge that this is an exploratory analysis, and further quantitative studies applying voxel-wise approaches are also required to evaluate the findings of the present work.

## 5. Conclusions

LvPPA displayed reduced ability to perform an adequate lexical retrieval and also some difficulties in sentence correctness that may deserve further exploration in future studies. Errors due to semantic disruption in the svPPA can be detected and quantified in narrative using the SDI, which showed a robust association with brain regions devoted to semantic memory.

## Figures and Tables

**Figure 1 brainsci-12-00910-f001:**
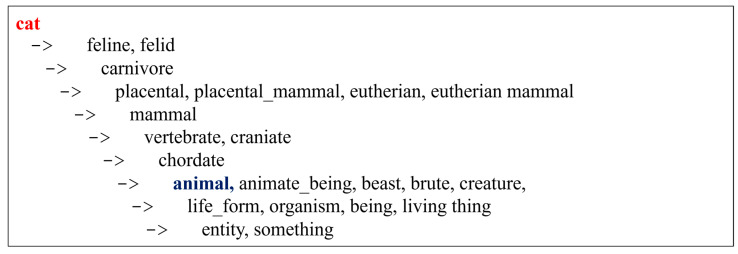
Complete hierarchical tree of the lexical entry “cat” as derived from WordNet. The Semantic Depth Index (SDI) of the specific term “cat” (in red) corresponds to 10, i.e., the number of nodes from the root term “entity” to the term “cat”. The SDI of the generic, superordinate term “animal” (in blue) corresponds to 3.

**Figure 2 brainsci-12-00910-f002:**
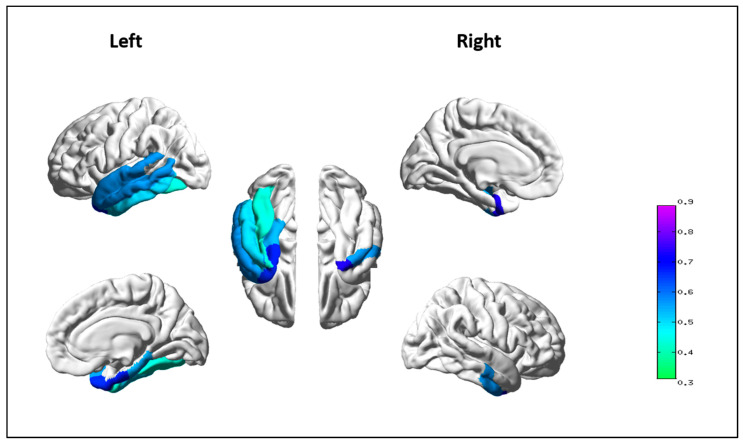
Brain regions correlating with the Semantic Depth Index in the whole sample of PPA patients (the color bar represents the value of Pearson’s r correlation coefficients).

**Table 1 brainsci-12-00910-t001:** Sociodemographic characteristics and neuropsychological assessment of the two clinical groups.

		svPPA		lvPPA	
**Sociodemographical variable**		M	SD	M	SD
Age (mean ± SD) years		72.50	8.044	74.44	7.519
Education (mean ± SD) years		13.17	5.565	8.94	4.449
Sex (M/F), *n*, %		3/3 (50%/50%)		8/8 (50%/50%)	
**Neuropsychological test**	**Cut-off**	M	SD	M	SD
Mini Mental State Examination	<23.80	13.92	5.92	17.20	6.78
Immediate recall of 15 Rey’s words	<28.53	13.86	5.64	22.89	8.48
Delayed recall of 15 Rey’s words	<4.69	1.96	3.51	3.38	2.76
Rey–Osterrieth figure copy	<28.87	22.85	15.46	14.13	9.20
Rey–Osterrieth figure recall	<9.46	3.75	3.43	3.73	4.81
Verbal span forward	<4.26	3.60	0.55	3.88	1.41
Verbal span backward	<2.65	3.00	1.87	2.38	1.20
Corsi’s test forward	<3.46	4.00	1.41	3.19	1.17
Corsi’s test backward	<3.08	3.40	0.89	2.69	1.14
Raven Colored Matrices	<18.96	23.75	9.98	17.55	5.81
Objet naming (oral)	<28	10.20	8.14	14.94	8.24

**Table 2 brainsci-12-00910-t002:** Means and standard deviations (SD) of linguistic features derived for both clinical groups. Abbreviations: CDA—conduites d’approche; SDI—semantic depth index; svPPA—patients with semantic variant of primary progressive aphasia; lvPPA—patients with logopenic variant of primary progressive aphasia; M—mean; SD—standard deviation.

	svPPA		lvPPA			
	M	SD	M	SD	U	*p*
**Number of nouns**	41.83	11.86	38.69	13.22	43.500	0.747
**Number of verbs**	53.33	8.73	36.38	12.23	12.500	**0.006**
**Number of adjectives**	17.83	8.42	7.13	3.90	14.000	**0.010**
**Number of open-class words**	285.83	78.04	161.81	55.09	8.000	**0.002**
**Number of close class words**	349.33	120.52	271.07	100.95	27.000	0.178
**Correct sentences**	118.00	26.07	56.06	24.90	4.000	**0.000**
**Phonemic paraphasias**	1.50	1.87	4.88	4.10	20.500	**0.040**
**Semantic paraphasias**	3.50	1.97	2.50	2.39	29.000	0.178
**Anomic pauses**	6.83	1.72	8.63	4.65	31.500	0.231
**CDA**	6.83	6.05	7.19	5.11	39.500	0.541
**Number of occurrences**	2.58	0.78	2.02	0.51	23.000	0.070
**SDI**	3.99	0.32	4.42	0.37	19.000	**0.033**
**Frequency of use**	689.14	105.38	768.25	209.85	34.000	0.329

**Table 3 brainsci-12-00910-t003:** Means and standard deviations of the cortical thickness of brain regions extracted using the Desikan’s atlas in the three groups (svPPA, lvPPA, and HC) and comparisons performed between each clinical group and healthy controls (HC) with Mann–Whitney U Tests. Abbreviations: SD—standard deviation; svPPA—semantic variant of Primary Progressive Aphasia; lvPPA—logopenic variant of Primary Progressive Aphasia. P-values were adjusted using the False Discovery Rate—(FDR) correction (Benjamini and Hochberg, 1995). In bold are reported statistically significant results.

			svPPA		lvPPA		HC		Pairwise Comparisons (pFDR)	
Lobe	Region	Hemisphere	Mean	SD	Mean	SD	Mean	SD	svPPA vs. HC	lvPPA vs. HC
Temporal lobe-medial aspect	Entorhinal cortex	Left	1.77	0.27	2.60	0.51	3.20	0.42	**<0.001**	**<0.001**
	Entorhinal cortex	Right	2.28	0.32	2.83	0.47	3.38	0.40	**<0.001**	**0.002**
	Fusiform gyrus	Left	1.99	0.25	2.18	0.43	2.56	0.11	**<0.001**	**<0.001**
	Fusiform gyrus	Right	2.15	0.22	2.30	0.18	2.57	0.11	**<0.001**	**<0.001**
	Parahippocampal gyrus	Left	1.88	0.24	2.25	0.46	2.58	0.26	**<0.001**	**0.005**
	Parahippocampal gyrus	Right	2.01	0.30	2.31	0.38	2.51	0.24	**0.010**	0.094
	Temporal pole	Left	2.34	0.51	3.06	0.59	3.47	0.39	**<0.001**	**0.037**
	Temporal pole	Right	2.83	0.72	3.17	0.53	3.59	0.35	**0.012**	**0.008**
Temporal lobe-lateral aspect	Banks superior temporal sulcus	Left	1.87	0.31	1.98	0.24	2.35	0.17	**0.005**	**<0.001**
	Banks superior temporal sulcus	Right	2.06	0.27	2.12	0.23	2.44	0.16	**0.005**	**<0.001**
	Inferior temporal gyrus	Left	1.99	0.31	2.30	0.37	2.66	0.19	**<0.001**	**<0.001**
	Inferior temporal gyrus	Right	2.36	0.25	2.43	0.23	2.70	0.17	**0.008**	**<0.001**
	Middle temporal gyrus	Left	2.04	0.33	2.34	0.35	2.70	0.18	**<0.001**	**<0.001**
	Middle temporal gyrus	Right	2.28	0.23	2.48	0.22	2.72	0.17	**0.005**	**<0.001**
	Superior temporal gyrus	Left	1.86	0.26	2.17	0.31	2.57	0.17	**<0.001**	**<0.001**
	Superior temporal gyrus	Right	2.16	0.26	2.34	0.21	2.60	0.19	**0.005**	**<0.001**
	Transverse temporal cortex	Left	1.97	0.37	1.92	0.34	2.21	0.25	0.191	**0.008**
	Transverse temporal cortex	Right	2.10	0.46	2.16	0.22	2.26	0.25	0.420	0.094
Frontal lobe	Caudal middle frontal gyrus	Left	2.14	0.26	2.05	0.25	2.40	0.16	**0.018**	**<0.001**
	Caudal middle frontal gyrus	Right	2.16	0.32	2.18	0.16	2.38	0.12	0.059	**<0.001**
	Frontal pole	Left	2.38	0.23	2.56	0.39	2.72	0.25	**0.018**	0.372
	Frontal pole	Right	2.53	0.21	2.70	0.23	2.76	0.29	0.071	0.461
	Lateral orbital frontal cortex	Left	2.27	0.30	2.50	0.34	2.57	0.15	**0.023**	0.981
	Lateral orbital frontal cortex	Right	2.38	0.19	2.51	0.22	2.52	0.16	0.103	0.873
	Medial orbital frontal cortex	Left	2.12	0.20	2.35	0.36	2.40	0.13	**0.008**	0.917
	Medial orbital frontal cortex	Right	2.28	0.21	2.45	0.19	2.41	0.19	0.199	0.524
	Paracentral lobule	Left	2.05	0.30	1.96	0.32	2.23	0.25	0.104	**0.004**
	Paracentral lobule	Right	2.15	0.19	2.03	0.21	2.23	0.23	0.338	**0.005**
	Pars opercularis	Left	2.22	0.26	2.21	0.24	2.45	0.13	**0.040**	**<0.001**
	Pars opercularis	Right	2.22	0.29	2.29	0.19	2.46	0.12	**0.043**	**0.005**
	Pars orbitalis	Left	2.34	0.35	2.43	0.32	2.54	0.14	0.199	0.476
	Pars orbitalis	Right	2.45	0.18	2.48	0.20	2.52	0.18	0.477	0.524
	Pars triangularis	Left	2.13	0.28	2.12	0.27	2.30	0.13	0.155	**0.025**
	Pars triangularis	Right	2.18	0.21	2.25	0.16	2.31	0.10	0.243	0.198
	Precentral gyrus	Left	2.13	0.30	2.04	0.30	2.38	0.24	0.063	**<0.001**
	Precentral gyrus	Right	2.20	0.27	2.11	0.26	2.35	0.23	0.220	**0.002**
	Rostral middle frontal gyrus	Left	2.01	0.25	2.07	0.24	2.27	0.11	**0.017**	**0.002**
	Rostral middle frontal gyrus	Right	2.09	0.22	2.15	0.13	2.25	0.12	0.061	**0.044**
	Superior frontal gyrus	Left	2.29	0.31	2.26	0.28	2.54	0.17	**0.049**	**<0.001**
	Superior frontal gyrus	Right	2.33	0.26	2.35	0.14	2.52	0.14	0.070	**<0.001**
Parietal lobe	Inferior parietal cortex	Left	1.93	0.27	1.91	0.28	2.29	0.13	**0.005**	**<0.001**
	Inferior parietal cortex	Right	2.09	0.20	2.01	0.20	2.32	0.14	**0.017**	**<0.001**
	Postcentral gyrus	Left	1.78	0.13	1.72	0.17	2.00	0.15	**0.008**	**<0.001**
	Postcentral gyrus	Right	1.81	0.17	1.82	0.13	1.97	0.18	0.059	**0.002**
	Precuneus cortex	Left	1.95	0.24	1.89	0.26	2.23	0.15	**0.012**	**<0.001**
	Precuneus cortex	Right	2.00	0.23	1.99	0.19	2.18	0.17	0.065	**0.007**
	Superior parietal cortex	Left	1.85	0.22	1.75	0.24	2.08	0.14	**0.008**	**<0.001**
	Superior parietal cortex	Right	1.92	0.24	1.84	0.16	2.07	0.16	0.092	**<0.001**
	Supramarginal gyrus	Left	2.13	0.24	1.97	0.22	2.38	0.13	**0.025**	**<0.001**
	Supramarginal gyrus	Right	2.21	0.16	2.08	0.12	2.36	0.17	0.050	**<0.001**
Occipital lobe	Cuneus cortex	Left	1.62	0.15	1.68	0.18	1.81	0.18	**0.022**	**0.021**
	Cuneus cortex	Right	1.64	0.15	1.66	0.13	1.75	0.20	0.211	0.063
	Lateral occipital cortex	Left	1.94	0.15	1.86	0.21	2.01	0.15	0.179	**0.008**
	Lateral occipital cortex	Right	1.97	0.20	1.96	0.14	2.09	0.16	0.220	**0.005**
	Lingual gyrus	Left	1.80	0.13	1.75	0.22	1.91	0.16	0.104	**0.017**
	Lingual gyrus	Right	1.82	0.14	1.81	0.10	1.90	0.16	0.234	**0.037**
	Pericalcarine cortex	Left	1.54	0.17	1.51	0.17	1.55	0.20	0.966	0.662
	Pericalcarine cortex	Right	1.50	0.17	1.50	0.18	1.57	0.18	0.350	0.179
Insulary lobe	Insula	Left	2.51	0.29	2.55	0.37	2.91	0.21	**0.012**	**<0.001**
	Insula	Right	2.69	0.13	2.68	0.29	2.92	0.20	**0.016**	**0.002**
Cingulate cortex	Caudal anterior-cingulate cortex	Left	2.61	0.19	2.58	0.37	2.68	0.29	0.390	0.782
	Caudal anterior-cingulate cortex	Right	2.53	0.14	2.57	0.28	2.63	0.28	0.343	0.654
	Isthmus–cingulate cortex	Left	2.01	0.11	1.97	0.27	2.29	0.24	**0.010**	**0.002**
	Isthmus–cingulate cortex	Right	2.11	0.13	1.99	0.16	2.26	0.17	0.092	**<0.001**
	Posterior-cingulate cortex	Left	2.09	0.24	2.11	0.34	2.40	0.20	**0.017**	**0.005**
	Posterior-cingulate cortex	Right	2.24	0.18	2.19	0.18	2.38	0.19	0.112	**0.008**
	Rostral anterior cingulate cortex	Left	2.39	0.32	2.55	0.41	2.80	0.25	**0.014**	**0.049**
	Rostral anterior cingulate cortex	Right	2.62	0.26	2.76	0.25	2.77	0.26	0.243	0.847

## Data Availability

The data presented in this study are available on request from the corresponding author.

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
