# Peer review of "Neuroanatomical Correlates of Semantic Features of Narrative Speech in Semantic and Logopenic Variants of Primary Progressive Aphasia"

_brainsci, 2022, doi:10.3390/brainsci12070910_

Round 1
Reviewer 1 Report
This research article by Quaranta et al focused on the correlation of the neuroanatomical changes with the semantic features of narrative speech in sv-PPA and lv-PPA. sv-PPA and lv-PPA patients were subjected to narrative speech tasks and a semantic depth index was determined. The patients also underwent brain MRI scanning. The authors found that the SDI of the patients correlate to their cortical thickness of the specific brain regions. Overall, the manuscript was organized and easy to follow and understand. I have only one minor comment: Please briefly describe the results section. No description was provided on the results shown in the tables/images (Tables 1-2, Figure 2).
Author Response
This research article by Quaranta et al focused on the correlation of the neuroanatomical changes with the semantic features of narrative speech in sv-PPA and lv-PPA. sv-PPA and lv-PPA patients were subjected to narrative speech tasks and a semantic depth index was determined. The patients also underwent brain MRI scanning. The authors found that the SDI of the patients correlate to their cortical thickness of the specific brain regions. Overall, the manuscript was organized and easy to follow and understand. I have only one minor comment: Please briefly describe the results section. No description was provided on the results shown in the tables/images (Tables 1-2, Figure 2).
R: We thank the reviewer. The results section now includes a more detailed description of the results.
Reviewer 2 Report
- Can authors introduce a table on semantic depth index values for two groups of progressive aphasias so the reader can figure out more closely the individual results
- Can the authors introduce some examples of results for both types of progressive aphasia subjects according to SDI? Similar to Fig 1, but so the reader can figure out how each response/error classified
- The Table in supplement can be added the manuscript not to be part of the supplement
- Last sentence in the Abstract (row 33-34) please rewrite it to be more understandable.
- Please use SDI acronym consistently in the paper (i.e. row 271, but check the entire manuscript)
- The results on cortical thickness could be enlarged in section 3.2 to put it in simple explanations, for example, cortical thickness was lower in scPPS vs lvPPA in the left entorhinal cortex…Please cover all the areas that showed to be significant compared to healthy control.
- The conclusion section could be more updated with study results for both progressive types since the title of the paper is related to both types.
- row 133 erase full stop after “Moreover.”
Author Response
Can authors introduce a table on semantic depth index values for two groups of progressive aphasias so the reader can figure out more closely the individual results
R: A supplementary table including the individual SDI scores of the patients has been added to the manuscript.
Can the authors introduce some examples of results for both types of progressive aphasia subjects according to SDI? Similar to Fig 1, but so the reader can figure out how each response/error classified
R: some examples have been added in the methods section.
-The Table in supplement can be added the manuscript not to be part of the supplement
R: The table has been revised to be easier to read and it has been added to the manuscript
-Last sentence in the Abstract (row 33-34) please rewrite it to be more understandable.
R: The sentence has been modified.
- Please use SDI acronym consistently in the paper (i.e. row 271, but check the entire manuscript)
R: The paper was revise accordingly.
- The results on cortical thickness could be enlarged in section 3.2 to put it in simple explanations, for example, cortical thickness was lower in scPPS vs lvPPA in the left entorhinal cortex…Please cover all the areas that showed to be significant compared to healthy control.
R: A more detailed description of the results concerning between groups comparisons on MRI measurements has been added to the results section.
- The conclusion section could be more updated with study results for both progressive types since the title of the paper is related to both types.
R: The conclusion has been updated.